# The CCR4–NOT Deadenylase Complex Maintains Adipocyte Identity

**DOI:** 10.3390/ijms20215274

**Published:** 2019-10-24

**Authors:** Akinori Takahashi, Shohei Takaoka, Shungo Kobori, Tomokazu Yamaguchi, Sara Ferwati, Keiji Kuba, Tadashi Yamamoto, Toru Suzuki

**Affiliations:** 1Cell Signal Unit, Okinawa Institute of Science and Technology Graduate University, Okinawa 904-0495, Japan; akinori.takahashi@oist.jp (A.T.); shohei.takaoka@oist.jp (S.T.); Sara.Ferwati@mail.mcgill.ca (S.F.); 2Nucleic Acid Chemistry and Engineering Unit, Okinawa Institute of Science and Technology Graduate University, Okinawa 904-0495, Japan; shungo.kobori@oist.jp; 3Depatment of Biochemistry and Metabolic Science, Graduate School of Medicine, Akita University, Akita 010-8543, Japan; yamaguchit@med.akita-u.ac.jp (T.Y.); kuba@med.akita-u.ac.jp (K.K.); 4Laboratory for Immunogenetics, Center for Integrative Medical Sciences, RIKEN, Kanagawa 230-0045, Japan

**Keywords:** ccr4–not complex, adipocyte, lipodystrophy

## Abstract

Shortening of poly(A) tails triggers mRNA degradation; hence, mRNA deadenylation regulates many biological events. In the present study, we generated mice lacking the *Cnot1* gene, which encodes an essential scaffold subunit of the CCR4–NOT deadenylase complex in adipose tissues (*Cnot1*-AKO mice) and we examined the role of CCR4–NOT in adipocyte function. *Cnot1*-AKO mice showed reduced masses of white adipose tissue (WAT) and brown adipose tissue (BAT), indicating abnormal organization and function of those tissues. Indeed, *Cnot1*-AKO mice showed hyperinsulinemia, hyperglycemia, insulin resistance, and glucose intolerance and they could not maintain a normal body temperature during cold exposure. Muscle-like fibrous material appeared in both WAT and BAT of *Cnot1*-AKO mice, suggesting the acquisition of non-adipose tissue characteristics. Gene expression analysis using RNA-sequencing (RNA-seq) showed that the levels of adipose tissue-related mRNAs, including those of metabolic genes, decreased, whereas the levels of inflammatory response-related mRNAs increased. These data suggest that the CCR4–NOT complex ensures proper adipose tissue function by maintaining adipocyte-specific mRNAs at appropriate levels and by simultaneously suppressing mRNAs that would impair adipocyte function if overexpressed.

## 1. Introduction

The adipose tissue regulates energy balance and lipid homeostasis [1]. White adipose tissue (WAT) provides energy storage, while brown adipose tissue (BAT) dissipates stored energy as heat [1]. Adipose tissue dysfunction or alterations in fat mass result in obesity (excess adipose tissue) or lipodystrophy (loss of adipose tissue). Although these two conditions are pathologically opposite, both are accompanied by similar metabolic abnormalities, including hepatic steatosis and insulin resistance, leading to increased risk of type 2 diabetes and cardiovascular disease [2,3]. Generalized and partial lipodystrophy are defined on the basis of the degree and location of fat loss [2]. Both genetic and acquired factors are responsible for these conditions [2]. Understanding the mechanism by which lipodystrophy develops will facilitate the development of therapeutic strategies.

Maintenance and expansion of adipocytes are also important for the homeostasis of the whole body besides that of the two types of adipose tissue [4]. Adipocytes are derived from multipotent mesenchymal stem cells, and various molecular events commit mesenchymal stem cells to the adipocyte lineage [4]. A complex network of transcription factors and cofactors is critically involved in adipocyte differentiation. Among them, the nuclear receptor peroxisome proliferator-activated receptor γ (PPARγ) and members of the CCAAT/enhancer-binding protein (C/EBP) family are central in adipogenesis [5].

While transcription participates in the control of gene expression and of biological events, the significance of mRNA decay in the regulation of mRNA abundance is becoming increasingly apparent [6,7]. Dysregulation of either mechanism leads to disorders, including cancer, neurodegenerative diseases, and diabetes. Deadenylation is an initial step in the decay of most mRNAs. Deadenylation facilitates the removal of the 5′ cap structure, leading to complete mRNA degradation mediated by Xrn1, a 5′–3′ exonuclease [8]. After removal of the poly(A) tail, mRNA is degraded from the 3′ end by the exosome complex, which contains 3′–5′ exonucleases [8]. The major deadenylase in mammals is the CCR4–NOT complex, which comprises at least eight subunits, i.e., CNOT1–3, CNOT6 (or CNOT6L), CNOT7 (or CNOT8), CNOT9–11 [9,10]. The two catalytic subunits, CNOT6/6L and CNOT7/8, belong to the exonuclease–endonuclease–phosphatase (EEP) family and the DEDD (Asp–Glu–Asp–Asp) family, respectively [11]. CNOT6/6L and CNOT7/8 have distinct biochemical functions in the complex [12,13]. CNOT1 is the scaffold protein of the complex and promotes mRNA deadenylation by facilitating the formation of the deadenylase complex [14,15,16,17]. Other non-catalytic subunits in the complex control deadenylase activity [18,19,20,21,22,23]. The subunits of the CCR4–NOT complex are ubiquitously expressed in adult mice with some tissue preferences, such as hematopoietic and metabolic tissues [24]. The CCR4–NOT complex deadenylates and consequently degrades mRNAs in a tissue-dependent manner, thereby regulating physiological functions of mammalian tissues, including energy metabolism, bone formation, B-cell development, adipocyte function, heart function, and liver maturation [20,25,26,27,28,29,30]. These findings underscore the physiological importance of CCR4–NOT complex-mediated mRNA decay. However, further analysis is necessary to understand the roles of its individual subunits in tissue function and homeostasis.

Here, we show that mice lacking the *Cnot1* gene (*Cnot1*-AKO mice) have reduced WAT and BAT masses, together with symptoms of lipodystrophy and cold sensitivity. Control and *Cnot1*-AKO mice differ significantly in the expression of mRNAs encoding molecules for adipose tissue function and inflammation, and these differences clearly influence the observed phenotypes. Therefore, CNOT1 suppression causes dysregulation of mRNA abundance, resulting in impaired adipocyte function. These findings provide further insights into mRNA deadenylation-mediated regulation of adipocytes.

## 2. Results

### 2.1. Suppression of CNOT1 in Mouse Adipose Tissues Leads to Reduced BAT and WAT Masses

While CNOT3, one of the non-catalytic subunits in the CCR4–NOT complex, is essential for various biological processes of mammalian tissues [20,26,27,28,29,30], the role of CNOT1, an essential scaffold subunit of the CCR4–NOT complex [14,15,16,17] in tissue function and homeostasis is not well known. To investigate CNOT1 functions in adipose tissues, we generated *Cnot1*-AKO mice by crossing adiponectin-cre mice with mice carrying a floxed allele for *Cnot1* (*Cnot1^loxP/loxP^*) and examined their phenotypes. CNOT1 protein largely disappeared in epididymal and inguinal WAT (eWAT and iWAT) as well as in BAT (Figure 1A). Control and *Cnot1*-AKO mice showed no obvious differences in appearance or body weight, although *Cnot1*-AKO mice showed a slightly increased food intake (Figure 1B–D). Masses of eWAT, iWAT, mesenteric WAT (mWAT), perirenal WAT (pWAT), and BAT were significantly lower in *Cnot1*-AKO mice than in control mice, whereas hepatic and pancreatic masses increased in *Cnot1*-AKO mice (Figure 1E,F). Therefore, the decrease in adipose tissue mass was offset by increased masses of liver and pancreas, resulting in similar total body weights between control and *Cnot1*-AKO mice. Muscle mass differed little between control and *Cnot1*-AKO mice (Figure 1F).

Histological analysis using hematoxylin and eosin (HE) staining revealed that lipid droplets were less abundant in BAT, iWAT, and eWAT of *Cnot1*-AKO mice (Figure 1G). Instead, lipids accumulated ectopically in the livers of *Cnot1*-AKO mice, contributing to the increased liver mass (Figure 1G). An increase of stromal cells, concomitant with a loss of adipocytes, was evident in eWAT of *Cnot1*-AKO mice (Figure 1G). Furthermore, muscle-like fibrous material was observed in iWAT and BAT of *Cnot1*-AKO mice (Figure 1G). Magnetic resonance imaging (MRI) confirmed a reduction of body fat content in *Cnot1*-AKO mice (Figure 1H). 

### 2.2. Cnot1-AKO Mice Display Lipodystrophy-Like Phenotypes

To address the relationship between reduced fat masses in *Cnot1*-AKO mice and adipocyte function, we examined glucose and lipid metabolism and found that the levels of circulating blood glucose, serum insulin, and serum triglycerides were significantly higher in *Cnot1*-AKO mice than in control mice (Figure 2A–C). Even after insulin administration, the blood glucose levels hardly decreased in *Cnot1*-AKO mice (Figure 2D). A glucose tolerance test showed that both control and *Cnot1*-AKO mice displayed glucose tolerance, as blood glucose levels dropped significantly, approximating those before glucose injection (Figure 2E). Glucose tolerance in *Cnot1*-AKO mice was unexpected, because *Cnot1*-AKO mice showed higher blood glucose levels under normal feeding conditions and insulin resistance (Figure 2A,D). *Cnot1*-AKO mice showed hyperinsulinemia (Figure 2B), which might partly explain the normal or slightly enhanced glucose metabolism (Figure 2E). Similar results have been reported in adipocyte-specific *Dicer*-KO mice [31]. Metabolic alterations, together with decreased adipocyte masses, indicate that adipose tissue-specific suppression of CNOT1 results in lipodystrophy-like phenotypes. 

### 2.3. Impaired BAT Function in Cnot1-AKO Mice

It is possible that an abnormal BAT structure, with fewer lipid droplets and muscle-like fibrous material, in *Cnot1*-AKO mice impairs thermogenesis, because BAT contributes to heat production, which is mediated by uncoupling protein 1 (UCP1) [32]. Body temperatures were similar between control and *Cnot1*-AKO mice when the mice were maintained at 22 °C (Figure 3A); however, body temperatures in *Cnot1*-AKO mice dropped below 20 °C during cold exposure (4 °C), whereas control mice maintained their temperatures above 35.0 °C (Figure 3A). Using immunoblots and immunohistochemistry, we detected decreased UCP1 in BAT of *Cnot1*-AKO mice (Figure 3B,C). Furthermore, oxygen consumption and energy expenditure in *Cnot1*-AKO mice did not change following treatments with CL316243, a β3-adrenergic receptor agonist that shows thermogenic anti-obese and insulin-sensitizing effects, while control mice significantly enhanced oxygen consumption and energy expenditure in the dark period (Figure 3D,E). Thus, *Cnot1*-AKO mice displayed phenotypes related to impaired BAT function.

### 2.4. Upregulation of mRNAs Irrelevant to Adipocyte Function and Downregulation of Metabolic Genes in Adipocytes Lacking CNOT1

To investigate the relationships between gene expression and observed abnormalities in *Cnot1*-AKO mice, we performed comprehensive mRNA profiling using RNA-seq results. We calculated fragments per kilobase of exon per million mapped sequence reads (FPKM) of genes (*Gene* FPKM). For comparison, we normalized *Gene* FPKMs with FPKM of *glyceraldehyde 3-phosphate dehydrogenase* (*Gapdh* FPKM), which was used as an internal control. We treated the normalized FPKMs as gene expression values. In iWAT of *Cnot1*-AKO mice, 877 mRNAs increased more than two-fold compared to those of control mice, and in BAT, 4063 mRNAs increased comparably (Figure 4A,B). Conversely, 1036 mRNAs in *Cnot1*-AKO iWAT, and 644 mRNAs in BAT decreased more than two-fold (Figure 4A,B). To characterize gene expression differences in iWAT and BAT, we performed gene ontology (GO) analysis of the lists of the upregulated and downregulated mRNAs. While functional annotation of upregulated mRNAs in iWAT of *Cnot1*-AKO mice showed enrichment of mRNAs that encode molecules involved in “neutrophil chemotaxis”, “chemotaxis”, and “cellular response to tumor necrosis factor”, downregulated mRNAs pertained mainly to enzymes involved in “oxidation–reduction process” and “lipid metabolic process” (Figure 4C and Appendix A). GO terms related to metabolism were also enriched among the downregulated mRNAs in BAT of *Cnot1*-AKO mice (Figure 4D and Appendix A). Many more mRNAs were upregulated in BAT than in iWAT of *Cnot1*-AKO mice (Figure 4A,B). Consequently, upregulated mRNAs showed more prominent enrichment of mRNAs relevant to immune responses (Figure 4D and Appendix A). Because lipodystrophy is associated with adipocyte inflammation [28,33], the results clearly explained the phenotype. We performed quantitative real-time PCR (qPCR) analysis and found that the levels of mRNAs relevant to adipocyte function were reduced in iWAT and BAT of *Cnot1*-AKO mice (Figure 4E). Those included mRNAs encoding a transcription factor involved in adipocyte differentiation and function (*Pparγ*) and a metabolic enzyme (*Pck1*). Similarly, immunoblot analysis showed a decrease in PPARγ, C/EBPα, C/EBPβ, and Perilipin, a lipid droplet formation-regulated protein (Figure 3C).

Since we observed muscle-like fibrous structure in iWAT and BAT of *Cnot1*-AKO mice (Figure 1G), we compared the expression of mRNAs and proteins relevant to muscle. qPCR and immunoblot analyses revealed that *Myod1* and *Myogenin* mRNAs and Myosin, Myogenin, and MyoD1 proteins increased in iWAT and BAT of *Cnot1*-AKO mice compared to control mice (Figure 3C and Figure 4E). Taken together, our findings suggest that CNOT1 helps to maintain adipose tissue homeostasis by regulating the abundances of mRNAs relevant to adipose tissues and by suppressing those mRNAs that impair adipocyte function if overexpressed.

### 2.5. Dysregulation of Unspliced mRNA Abundance Contributes to Gene Expression Differences in Adipose Tissues between Control and Cnot1-AKO Mice

Suppression of the CCR4–NOT complex is generally relevant to mRNA stabilization in mouse tissues [20,25,26,27,29,30]. We performed chase experiments to examine mRNA stability in adipocytes. We prepared primary mature adipocytes from iWAT and BAT of control and *Cnot1*-AKO mice and treated them with the transcription inhibitor actinomycin D (Act. D). Total RNA was prepared from Act. D-treated adipocytes and subjected to RNA-seq. The results of comprehensive mRNA half-life profiling showed that there were more destabilized (half-life *Cnot1*-AKO/control ratio <0.5) than stabilized (half-life *Cnot1*-AKO/control ratio >2.0) mRNAs in both iWAT and BAT of *Cnot1*-AKO mice (Figure 5A,B). While enriched Biological Process (BP) GO terms of destabilized mRNAs were similar to those of downregulated mRNAs in both iWAT and BAT, there was no similarity in enriched BP GO terms between stabilized and upregulated mRNAs in iWAT (Figure 5C,D and Appendix A). In addition, no GO terms were specifically enriched among the stabilized mRNAs in BAT. These findings suggest that increased levels of mRNAs in adipocytes of *Cnot1*-AKO mice were probably little impacted by mRNA stabilization. On the other hand, mRNA destabilization decreased mRNA levels in both iWAT and BAT of *Cnot1*-AKO mice.

We next compared the expression of unspliced, premature mRNAs (pre-mRNAs). We counted the number of intron sequence reads (intronic FPKM) using the RNA-seq results. In iWAT of *Cnot1*-AKO mice, intronic FPKMs of 576 genes increased and those of 911 decreased more than two-fold compared to control mice (Figure 6A). Moreover, enriched GO terms of genes showing decreased intronic FPKMs were similar to those of decreased and destabilized mRNAs (Figure 4B, Figure 5C and Figure 6C). On the other hand, in BAT, the difference in intronic FPKMs between control and *Cnot1*-AKO mice was comparable to that of mRNA expression (Figure 4B and Figure 6B). Consistent with this, enriched GO terms of genes showing increased or decreased intronic FPKMs were similar to those of upregulated or downregulated mRNAs, respectively, in BAT of *Cnot1*-AKO mice (Figure 4D and Figure 6D). In addition to direct compare the enriched GO terms relative to mRNA and pre-mRNA expression, we performed GO analyses using lists of genes that commonly increase or decrease at both mRNA and pre-mRNA levels in adipocytes of *Cnot1*-AKO mice. We detected similar enrichment of GO terms (Appendix A). These findings suggest that in adipose tissues, changes in pre-mRNA levels influence differences in gene expression between control and *Cnot1*-AKO mice more significantly than does mRNA stability.

## 3. Discussion

The CCR4–NOT complex serves essential functions in the development and homeostasis of various mammalian tissues [20,25,26,27,28,29,30]. In murine adipose tissues, the complex prevents disease-related phenotypes, such as obesity and lipodystrophy [25,28]. Suppression of the CCR4–NOT complex leads to stabilization of mRNAs in various tissues, and these stabilized mRNAs are responsible for functional abnormalities. In this study, we found that in adipocytes, mRNAs were stabilized upon CCR4–NOT complex suppression, but mRNA stabilization was not the primary factor causing dysregulated gene expression. Compared to the number of upregulated mRNAs in BAT of *Cnot1-*AKO mice, a much smaller number of mRNAs showed stabilization, and many more mRNAs were destabilized (Figure 4B and Figure 5B). Increased pre-mRNA levels accounted for most of the increased expression of mRNAs (Figure 6B,D). Similarly, mRNA destabilization and changes in pre-mRNA expression were more responsible for dysregulated gene expression than mRNA stabilization in iWAT of *Cnot1*-AKO mice (Figure 4, Figure 5 and Figure 6). Pre-mRNA expression is used as a proxy for transcription rate [34], suggesting that changes in transcriptional programs contribute to differences in gene expression in adipose tissues between control and *Cnot1*-AKO mice. Previous studies have shown that the CCR4–NOT complex directly regulates transcription in mammals [35,36,37,38,39], suggesting a critical interaction between the CCR4–NOT complex and the transcription machinery directly or indirectly in murine adipocytes. Importantly, in yeast, other mRNA decay factors in addition to the CCR4–NOT complex, shuttle between the cytoplasm and the nucleus and regulate transcription in a manner dependent on mRNA degradation [6]. It is also possible that the decay factors that act subsequent to deadenylation, such as XRN1 and DCP1 [8], directly influence transcription in adipose tissues.

microRNAs (miRNAs) are involved in adipogenesis and adipocyte function [40,41,42,43]. In mice, lack of *Dicer1*, an enzyme for miRNA processing, results in loss of adipocyte identity [31]. We showed that mice lacking subunits of the CCR4–NOT complex had similar adipocyte abnormalities, including development of lipodystrophy and cold sensitivity ([28] and this study). While miRNAs control tissue-specific transcriptional networks to maintain tissue identity [44,45], dysregulated transcriptional programs were observed in tissues, including adipose tissues, upon suppression of the CCR4–NOT complex ([27,29,30] and this study). Therefore, both mechanisms are essential to fine-tune transcriptional programs for the maintenance of tissue identity. It is also possible that the CCR4–NOT complex cooperates with the miRNA pathway, because the CCR4–NOT complex uses the miRNA–Argonaute complex to recognize some target mRNAs [46].

We observed a muscle-like fibrous structure in BAT and iWAT of *Cnot1*-AKO mice in addition to reduced adipose tissue mass (Figure 1). Mesenchymal stem cell differentiation into muscle or adipose tissue is mediated mainly by MyoD or PPARγ, respectively [47,48]. Importantly, MyoD-mediated muscle differentiation and PPARγ-mediated adipose tissue differentiation are mutually exclusive [49]. Moreover, the decrease of PPARγ in the adipose tissues of *Cnot1*-AKO mice was concomitant with the increase of MyoD (Figure 3C). These data suggest that the loss of adipose tissue characteristics, at least in part, leads to the induction of fibrous material in adipose tissues. Intriguingly, muscular hypertrophy is observed in patients with both generalized and partial lipodystrophy [50,51,52]. Further analyses are required to determine whether the antagonism between adipose tissue and muscle differentiation is responsible for the appearance of this fibrous material in the adipose tissues lacking CNOT1 and for muscular hypertrophy in lipodystrophy. *Cnot1*-AKO mice could be a useful animal model to examine these phenomena.

In conclusion, our findings show that CCR4–NOT complex-mediated regulation of gene expression has significant effects on proper adipocyte function, including insulin sensitivity, lipid metabolism, and control of body temperature. The appearance of other tissue characteristics clearly indicates loss of adipocyte identity. These findings raise the possibility of developing therapeutic strategies to treat metabolic defects and lipodystrophy by supporting the function of the CCR4–NOT complex in the adipose tissue.

## 4. Materials and Methods 

### 4.1. Mice

The creation of *Cnot1* conditional knockout (*Cnot1^loxP/loxP^*) mice has been previously described [29]. We backcrossed *Cnot1^loxP/loxP^* mice with C57BL/6J mice for at least eight generations. Mice were maintained on a 12 h light/dark cycle in a temperature-controlled (22 °C) barrier facility with free access to water and a normal diet (NCD, CA-1, CLEA Japan, Inc., Meguro, Tokyo, Japan). To generate adipose tissue-specific *Cnot1*-AKO mice, we crossed *Cnot1^loxP/loxP^* mice with *Adiponectin-Cre* mice (The Jackson Laboratory, Stock No: 010803, Bar Harbor, ME, USA). We used *Cnot1^loxP/loxP^* mice as controls unless otherwise indicated. Mouse experiments were approved by the Committee of Animal Experiments at Okinawa Institute of Science and Technology Graduate University (2015-122, 7/12/2015; 2012-054, 1/12/2012).

### 4.2. Antibodies

The anti-CNOT1 mouse monoclonal antibody was described previously [25]. Antibodies against GAPDH (#2118), PPARγ (#2443), C/EBPβ (#3087), and Perilipin (#9349) were purchased from Cell Signaling Technology (Danvers, MA, USA). The antibody against C/EBPα (sc-61) was from Santa Cruz Biotechnology (Dallas, TX, USA). Antibodies against UCP1 (ab10983), MyoD1 (ab16148), and Myogenin (ab124800) were from Abcam (Cambridge, UK). The antibody against Myosin (M4276) was from Sigma Aldrich (St. Louis, MO, USA).

### 4.3. Cell Culture

Primary mature adipocytes were obtained from iWAT and BAT of 12-week-old mice. The adipose tissue was minced and digested in Hanks’ Balanced Salt Solution (HBSS; Thermo Fisher Scientific, Wilmington, MA, USA) containing 1.5 U/mL collagenase D (Roche, Basel, Switzerland) and 2.4 U/mL dispase II (Roche) at 37 °C for 45 min. The digestions were stopped with DMEM/F-12 containing 10% FBS. The cells were filtered through a 100 µm cell strainer (CORNING, Corning, New York, USA) and centrifuged at 250× *g* for 5 min. The cells in the supernatant were cultured in DMEM/F-12 containing 10% FBS.

### 4.4. GTT and ITT

For the glucose tolerance test (GTT), mice were fasted for 24 h. We injected mice intraperitoneally with 0.5 mg of D(+)-glucose per g body weight. For the insulin tolerance test (ITT), we injected mice intraperitoneally with 1.0 mU of human insulin (Eli Lilly and Co., Indianapolis, IN, USA) per g body weight. We collected blood samples and measured glucose concentrations with a glucometer (Glutest Pro, Sanwa Kagaku Kenkyusho, Nagoya, Aichi, Japan). Insulin concentration was determined with a mouse insulin ELISA kit (Mercodia, Uppsala, Sweden), and triglyceride concentration was measured with the Triglyceride E-Test (Wako, Osaka, Japan) according to the manufacturer’s protocols.

### 4.5. Cold Tolerance Test and Metabolic Analyses

Mice were individually housed at 4 °C for 2 h. Core body temperature was monitored every hour with a Microprobe Thermometer (BAT-12; Physitemp, Clifton, NJ, USA). Oxygen consumption and energy expenditure were measured with an Oxymax system (Columbus Instruments, Columbus, OH, USA). For β3-agonist stimulation, the mice were intraperitoneally injected with 1 µg/g body weight of CL316243 (Sigma Aldrich). 

### 4.6. Magnetic Resonance Imaging (MRI) 

The mice were scanned using a 11.7 T Bruker MRI system under isoflurane anesthesia. For each mouse, the whole body was imaged in accordance with an MRI fat protocol. Parameters for short T1-weighted spin-echo pulse sequences were: repetition time = 360 ms, echo time = 20 ms, slice thickness = 1.0 mm, field of view = 3.5 × 3.5 (cm^2^), matrix size = 320 × 320, average = 8. A fat image region was evaluated by visual inspection.

### 4.7. Histological Analysis of Tissues

After dissection, iWAT, BAT, and liver were fixed in 10% formaldehyde overnight and embedded in paraffin. Paraffin-embedded sections were stained with Hematoxylin 3G (8656) and Eosin (8659) from Sakura Finetek (Chuo, Tokyo, Japan). Immunohistochemistry for UCP1 was performed with an antibody against UCP1, as described previously [30].

### 4.8. RNA sequencing

Total RNA was isolated from iWAT and BAT of mice at 12 weeks of age using Isogen II (Nippongene, Chiyoda, Tokyo, Japan). For comprehensive mRNA half-life profiling, mature adipocytes from iWAT and BAT were treated with 2.5 µg/mL actinomycin D (Wako). Total RNA was prepared at 0, 4, and 8 h after treatment. Total RNA (1 µg) was used for RNA-seq library preparation with TruSeq Stranded mRNA LT Sample Prep Kit (Illumina, San Diego, CA, USA), according to the manufacturer’s protocol. We performed 109 base-pair paired-end RNA sequencing with a Hiseq PE Rapid Cluster Kit v2-HS and a Hiseq Rapid SBS Kit v2-HS (200 Cycle) on a Hiseq2500 (Illumina), according to the manufacturer’s protocol. The reads were mapped to the Ensembl genome sequence with StrandNGS (Strand Life Sciences, Bangalore, India). For data analysis, using StrandNGS software (Strand Life Sciences), the reads were mapped to the Ensembl genome sequence (mm10), and FPKMs were calculated. Protein-coding genes were selected for comparison. Genes with <0.1 FPKM were excluded. Gene expression values represent *Gene* FPKMs normalized with *Gapdh* FPKM. For calculation of mRNA half-lives, the intercept and slope of the linear regression line were used according to the formula: LN(0.5/e^intercept)/slope. mRNAs with <0.1 FPKM and half-lives less than 0 h or longer than 50 h were excluded. For calculating intronic FPKM (iFPKM), reads mapped to the intronic region were extracted with StrandNGS software, and read numbers were normalized by the total count of reads mapped to the intronic region and the sum of intron lengths. Genes with FPKM < 0.1, iFPKM < 0.001, and intron length < 150 bp were eliminated. Sequence data are available through ArrayExpress under the accession number (E-MTAB-8366 and E-MTAB-8371).

### 4.9. Quantitative Real-Time RT-PCR

Total RNA (1 µg) was reverse-transcribed to cDNA with SuperScript Reverse Transcriptase III (Thermo Fisher Scientific) as described previously [25]. The cDNA was mixed with primers and SYBR Green Supermix (Takara, Kusatsu, Shiga, Japan) and analyzed with a Viia 7 sequence detection system (Thermo Fisher Scientific). The relative expression of mRNA was determined after normalization to the *Gapdh* level using the ∆∆Ct method. Primers are listed in Appendix A.

### 4.10. Immunoblot Analyses

Immunoblotting was performed as described previously [25]. iWAT and BAT were solubilized in TNE buffer (50 mM Tris-HCl (pH 7.5), 150 mM NaCl, 1 mM EDTA, 1% NP40, and 1 mM PMSF) for 30 min at 4 °C. Lysates in SDS sample buffer were subjected to SDS-polyacrylamide gel electrophoresis and electro-transferred onto Immobilon-P membranes (Millipore, Burlington, MA, USA). Protein bands were detected with primary antibodies and ECL anti-rabbit or mouse IgG Horseradish Peroxidase (HRP)-linked whole antibodies (GE Healthcare, Chicago, IL, USA) as the secondary antibodies. For detection, we used Immobilon Western HRP substrate (Millipore). We analyzed the results using ImageQuant software and an Image Analyzer LAS 4000 mini (GE Healthcare).

### 4.11. Bioinformatic Analysis

GO enrichment analysis was performed with DAVID Bioinformatics Resources 6.8 (https://david.ncifcrf.gov). 

### 4.12. Statistical Analyses

We used unpaired, two-tailed Student’s t tests. Values represent means ± standard error of the means (sem) and are represented as error bars. A *p*-value <0.05 was considered statistically significant.

## Figures and Tables

**Figure 1 ijms-20-05274-f001:**
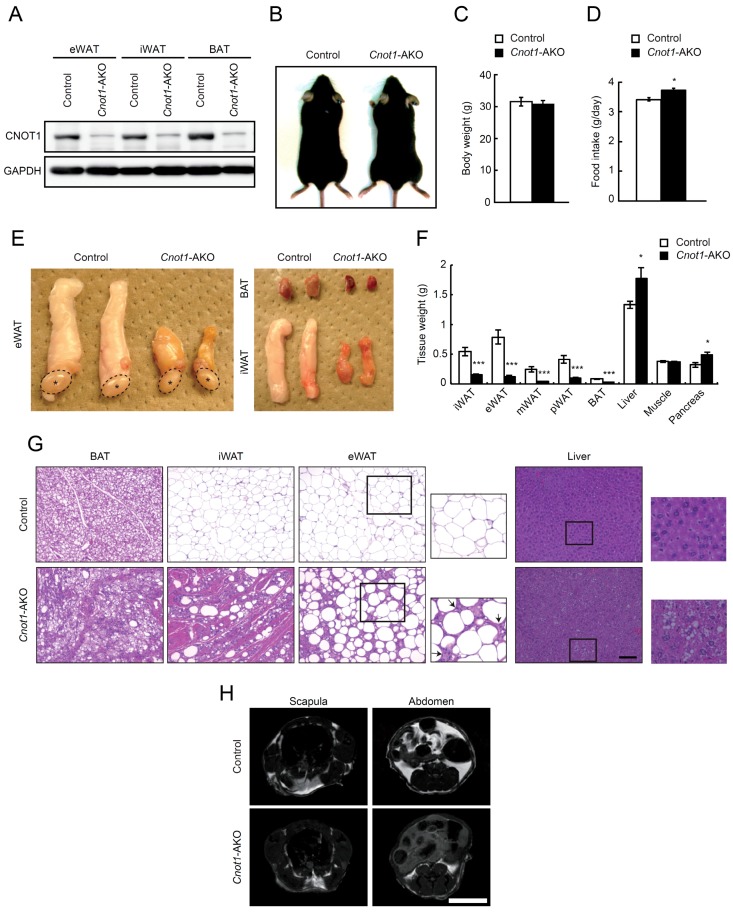
Mice lacking *Cnot1* in adipose tissues display fewer adipocytes. (**A**) Immunoblotting for CNOT1 and GAPDH in epididymal white adipose tissue (eWAT), inguinal WAT (iWAT), and brown adipose tissue (BAT) of control and *Cnot1*-AKO mice at 32 weeks of age. (**B**) Gross appearance of control and *Cnot1*-AKO mice. (**C**) Body weights of control and *Cnot1*-AKO mice at 32 weeks of age (*n* = 8). (**D**) Food intake of 28-week-old control (*n* = 4) and *Cnot1*-AKO (*n* = 3) mice. (**E**) Gross appearance of eWAT, iWAT, and BAT in 28-week-old mice. Asterisks indicate the testes. (**F**) Tissue weights of control and *Cnot1*-AKO mice at 32 weeks of age (*n* = 8). (**G**) Hematoxylin and eosin (HE) staining of BAT, iWAT, eWAT in 32-week-old and of the liver in 29-week-old mice. Magnified images (rectangles) of eWATs and livers are shown in righthand panels. Arrows indicate an increase of stromal cells. Scale bar, 100 µm. (**H**) MRI evaluation of fat accumulation in 20-week-old mice. Scale bar, 1 cm. Values in graphs represent mean ± sem. * *p* < 0.05, ** *p* < 0.01, *** *p* < 0.001, unpaired t-test.

**Figure 2 ijms-20-05274-f002:**
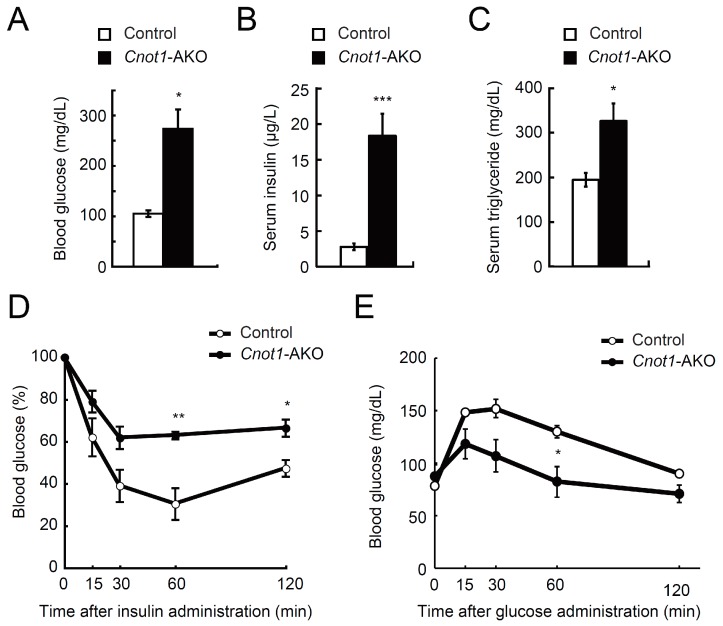
CNOT1 suppression in adipose tissues causes generalized lipodystrophy. (**A**) Levels of blood glucose in 22-week-old control (*n* = 3) and *Cnot1*-AKO mice (*n* = 4) under feeding conditions. (**B**,**C**) Levels of serum insulin (**B**) and serum triglycerides (**C**) in 22-week-old mice (*n* = 3) under feeding conditions. (**D**) Insulin tolerance test (ITT) in 28-week-old control (*n* = 3) and *Cnot1*-AKO mice (*n* = 4). (**E**) Glucose tolerance test (GTT) in 28-week-old control (*n* = 3) and *Cnot1*-AKO mice (*n* = 4). Values in graphs represent mean ± sem. * *p* < 0.05, ** *p* < 0.01, *** *p* < 0.001, unpaired t-test.

**Figure 3 ijms-20-05274-f003:**
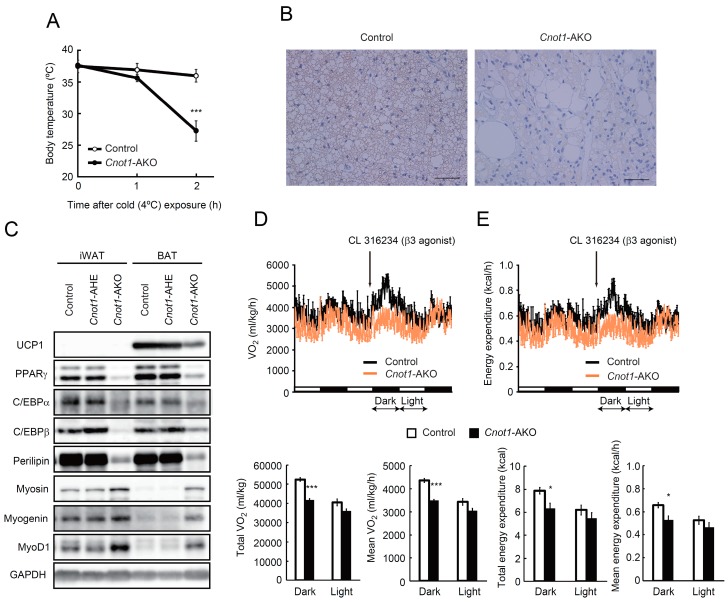
*Cnot1*-AKO mice show cold intolerance. (**A**) Cold tolerance test in 20-week-old control (*n* = 5) and *Cnot1*-AKO (*n* = 4) mice, individually housed at 4 °C for 2 h. The core body temperature was monitored every hour with a thermometer. (**B**) Immunohistochemistry for UCP1 in BAT of control and *Cnot1*-AKO mice. Scale bars, 50 µm. (**C**) Immunoblot analyses of the indicated proteins in BAT of control, *Cnot1*-AHE, and *Cnot1*-AKO mice. *Cnot1*-AHE indicates mice with the genotype *Cnot1^+/loxP^*:*Adiponectin-cre*. (**D**,**E**) Oxygen consumption (**D**) and energy expenditure (**E**) in 20-week-old mice were measured. Mice were injected intraperitoneally with CL316243 at the indicated time points (arrows) (*n* = 3). Both the total and the mean oxygen consumption and energy expenditure were calculated separately for dark and light periods after β3-agonist injection (12 h each, double arrows), and are shown in bar plots. Values in the graphs represent mean ± sem. * *p* < 0.05, *** *p* < 0.001, unpaired t-test.

**Figure 4 ijms-20-05274-f004:**
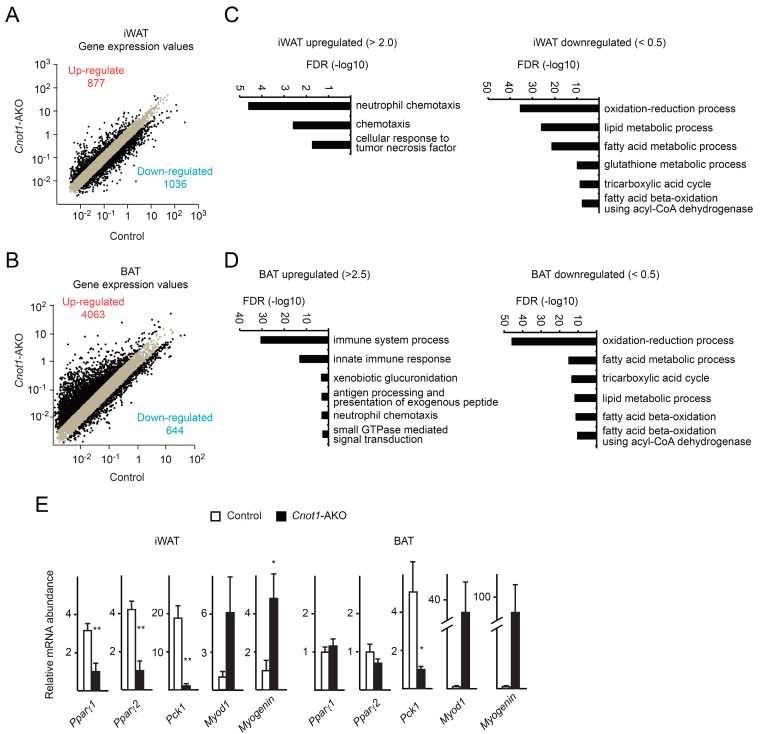
CNOT1 suppression in adipocytes leads to an increase of mRNAs for immunoresponse and a decrease of those for metabolic processes. (**A**,**B**) Scatter plots of gene expression values in iWAT (**A**) and BAT (**B**) of 12-week-old control and *Cnot1*-AKO mice (*n* = 4). Black dots represent mRNAs whose expression differed more than two-fold in *Cnot1*-AKO mice compared to controls. (**C**,**D**) (GO) enrichment analysis of mRNAs that were upregulated (>2.0 in iWAT and >2.5 in BAT) or downregulated (<0.5) in iWAT (**C**) and BAT (**D**) of 12-week-old *Cnot1*-AKO mice. Bar charts of the Biological Process (BP) GO terms (at most six) ranked by the false discovery rate (FDR) (<0.05) are shown. Gene lists included in each GO term are summarized in Appendix A. (**E**) Quantitative PCR analysis of the indicated mRNA levels in iWAT and BAT of 12-week-old mice (*n* = 4). Values in the graphs represent mean ± sem. * *p* < 0.05, ** *p* < 0.01, *** *p* < 0.001, unpaired t-test.

**Figure 5 ijms-20-05274-f005:**
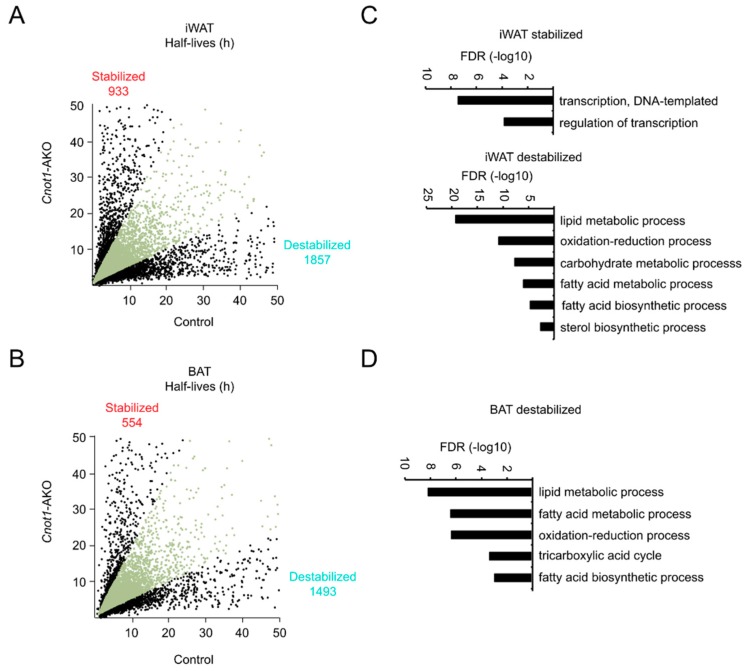
mRNA destabilization is more prominent than mRNA stabilization in adipose tissues of *Cnot1*-AKO mice. (**A**,**B**) Scatter plots of mRNA half-lives in iWAT (**A**) and BAT (**B**) of 12-week-old control and *Cnot1*-AKO mice (*n* = 4). Black dots represent mRNAs with half-lives more than two-fold longer or shorter in *Cnot1*-AKO mice than in controls. (**C**,**D**) GO enrichment analysis of genes that were stabilized (half-life *Cnot1*-AKO/control ratio >2.0) or destabilized (half-life *Cnot1*-AKO/control ratio <0.5) in iWAT (**C**) and BAT (**D**) of 12-week-old *Cnot1*-AKO mice. Bar charts of the BP GO terms (at most six) ranked by FDR (<0.05) are shown. Note that no significant enrichment of GO terms occurred in stabilized mRNAs in BAT. Gene lists included in each GO term are summarized in Appendix A.

**Figure 6 ijms-20-05274-f006:**
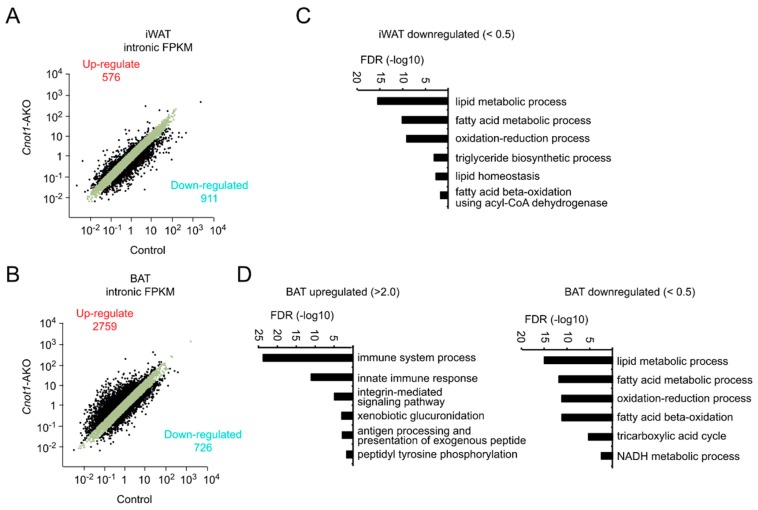
Changes in pre-mRNA abundances are responsible for gene expression differences between control and *Cnot1*-AKO mice. (**A**,**B**) Scatter plots of intronic fragments per kilobase of exon per million mapped sequence reads (FPKMs) in iWAT (**A**) and BAT (**B**) of 12-week-old control and *Cnot1*-AKO mice (*n* = 4). Black dots represent pre-mRNAs whose expression differed more than two-fold in *Cnot1*-AKO mice compared to controls. (**C**,**D**) GO enrichment analysis of genes showing intronic FPKMs that increased or decreased more than two-fold in iWAT (**C**) and BAT (**D**) of 12-week-old *Cnot1*-AKO mice. Bar charts of the BP GO terms (at most six) ranked by FDR (<0.05) are shown. Note that no significant enrichment of the BP GO terms occurred for increased pre-mRNAs in iWAT. Gene lists included in each GO term are summarized in Appendix A.

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
