# Peer review of "The CCR4–NOT Deadenylase Complex Maintains Adipocyte Identity"

_ijms, 2019, doi:10.3390/ijms20215274_

Round 1

Reviewer 1 Report

The manuscript entitled “The CCR4-NOT deadenylase complex maintains adipocyte identity ” by  Takahashi et al. investigates the role of the CCR4-NOT deadenylase complex in adipocyte function by generating an adipose tissue specific CNOT1 knockout mouse. CNOT1 depleted mice show decreased BAT and WAT mass, hyperinsulememia, hyperglycaemia, insulin resistance and glucose as well as cold intolerance.  Muscle-fibrous material appeared in both WAT and BAT of the ko mice, suggesting acquisition of non-adipose characteristics. Transcriptome analysis revealed decreased expression of adipose and metabolic related genes and increase of inflammatory response related mRNAs levels. All in all, the data presented in this manuscript indicate that CCR4-NOT complex play a key role in adipose tissue function ensuring appropriate expression of adipose specific genes and suppressing mRNAs that would impair adipose tissue function if upregulated.

The novelty of the paper consists in the identification of the CCR4-NOT complex as an important player in the maintenance adipose tissue function.

The way the manuscript has been written is clear, however some revision is required to make the manuscript suitable to be published in IJMS.

Major comments

Results:

Figure 1F. Since the total body weight is similar between wt and ko mice, the increased weight of liver and pancreas of the knockout mouse account for the decrease in the weight of the adipose tissue depots? Can the authors comment on that.

Figure 1G. In the text of the manuscript the authors mention a crown-like structure in the histology of the WAT, but in the figure panel that is not indicated. Can the authors indicate this structure on the histology figure?

The authors also mention increased fat deposition in the liver of the ko mice vs controls, but looking at the histology of the livers of the two mouse models appear quite similar. The authors should provide a more representative image of the steatotic liver of the ko mice, if they claim that there is more fat deposition in the liver of that mouse model.

Figure 2A-C. The insulin, glucose and TG maesurements have been done in blood of fasted or fed mice?

Figure 2D. The wt mice do not respond to the glucose injection in the GTT showed in this panel. The authors should provide a better control.

Figure 3D. Is the oxygen consumption in the ko mice significantly reduced vs controls? The authors should provide statistical evidence there.

Figure 4F. The authors should provide the loading control (GAPDH) for this blott too. The fact that the lysate used were the same that the ones showed in Figure 3 does not count is the membranes were not the same.

Minor comments

Results

Figure 3C.The loading control GAPDH is overexposed. The authors should provide a better blott.

Reviewer 2 Report

The manuscript by Takahashi and co-authors focus on the study of the role of Cnot1-A (a subunit of the CCR4-NOT deadenylase complex of mRNA in adipose tissue).

The manuscript is very interesting and well organized. Results are clear and in line with the aim. Moreover, they represent the basis for understanding the mechanism leading adipose tissue dysfunctions.

I just suggest the authors improve the discussion section.

Round 2

Reviewer 1 Report

I am satisfied of how the authors addressed my comments a part of two of them:

1)Figure 1G. In the text of the manuscript the authors mention a crown-like structure in the histology of the WAT, but in the figure panel that is not indicated. Can the authors indicate this structure on the histology figure?

The authors indicated in the figure the crown-like structures they meant in the text, but the resolution of the figure does not allow to identify the actual structure. Or anther explanation can be that the structure is not there at all. Can the authors usa a marker of crown-like structures to help identifying them or provide a better quality/representative image?

2)Figure 2D. The wt mice do not respond to the glucose injection in the GTT showed in this panel. The authors should provide a better control.

The authors in this case did change the n number of replicates in the legend but the graph is still the same. The wt mice still do not show a response to the GTT, what does not represent the correct control. Or the authors forgot to replace the figure panel or they did not repeat the experiment. In both cases that does not address appropriately my comment.
